# Sign Language Avatars: A Question of Representation

**Rosalee Wolfe** [1,*], **John C. McDonald** [2], **Thomas Hanke** [3], **Sarah Ebling** [4], **Davy Van Landuyt** [5], **Frankie Picron** [5], **Verena Krausneker** [6], **Eleni Efthimiou** [1], **Evita Fotinea** [1] and **Annelies Braffort** [7]

1. Institute of Language and Speech Processing, Athena Research and Innovation Center, GR-151 25 Maroussi, Greece; eleni_e@athenarc.gr (E.E.); evita@athenarc.gr (E.F.)
2. School of Computing, DePaul University, 243 S. Wabash Ave., Chicago, IL 60604, USA; jmcdonald@cs.depaul.edu
3. Institut für Deutsche Gebärdensprache, Universität Hamburg, 20354 Hamburg, Germany; thomas.hanke@uni-hamburg.de
4. Department of Computational Linguistics, University of Zurich, 8050 Zurich, Switzerland; ebling@cl.uzh.ch
5. European Union of the Deaf, Rue de la Loi-Wetstraat 26/15, 1040 Brussels, Belgium; davy.van.landuyt@eud.eu (D.V.L.); frankie.picron@eud.eu (F.P.)
6. Institut für Bildungswissenschaft, Universität Wien, Liebiggasse 5, 1010 Vienna, Austria; verena.krausneker@univie.ac.at
7. Laboratoire Interdisciplinaire des Sciences du Numérique, Campus universitaire bât 507, Rue du Belvedère, F-91405 Orsay, France; annelies.braffort@lisn.upsaclay.fr
* Correspondence: rosalee.wolfe@athenarc.gr

**Abstract:** Given the achievements in automatically translating text from one language to another, one would expect to see similar advancements in translating between signed and spoken languages. However, progress in this effort has lagged in comparison. Typically, machine translation consists of processing text from one language to produce text in another. Because signed languages have no generally-accepted written form, translating spoken to signed language requires the additional step of displaying the language visually as animation through the use of a three-dimensional (3D) virtual human commonly known as an avatar. Researchers have been grappling with this problem for over twenty years, and it is still an open question. With the goal of developing a deeper understanding of the challenges posed by this question, this article gives a summary overview of the unique aspects of signed languages, briefly surveys the technology underlying avatars and performs an in-depth analysis of the features in a textual representation for avatar display. It concludes with a comparison of these features and makes observations about future research directions.

**Keywords:** machine translation; sign language translation; sign language representation; avatar technology; sign language linguistics; computer animation

## 1. Introduction

An essential feature of any automatic translation system is the ability to display the target language in a manner that is easy to understand. The language produced should be grammatically and idiomatically correct. Researchers have made significant progress in translating between high-resource spoken languages that have a written form, and some have even suggested that automatic translation has achieved human parity in some domains [1].

Given the progress in translating text from one spoken language to another, one would expect to see similar advancements in addressing the question of translating between spoken and signed languages. However, progress in the translation between the two has lagged significantly in comparison. Traditionally, the process of machine translation is typically viewed as converting text from one language to the text of another. Since signed languages have no widely accepted written form, an additional required step is displaying signed languages in their natural moving form, in the visual modality [2].

Although research efforts have investigated this problem for over twenty years [3] it is still an open question. With the goal of developing a deeper understanding of the process of converting a textual to a visual language, this article gives a brief overview of the unique aspects of signed languages, reviews the technology underlying avatars and identifies representation features useful for avatar display. It concludes with a comparison of these features and makes observations about future research directions.

### 1.1. Background

Signed languages are distinct from the spoken languages surrounding them. For example, in France, many deaf persons use Langue des Signes Française (LSF), not French, as their preferred language. Since French is a second language to them, even its written form poses a barrier. Many researchers have noted that written language poses barriers to members of Deaf communities [4–7].

Deaf signed language users consider themselves members of a minority group, with a distinct language, culture, and shared experiences, rather than as people with a disability [8]. They continually struggle with the reality that policymakers in such institutions as governmental departments, educational institutions and health care agencies consist primarily of hearing people who are not familiar with the values, goals, and concerns of signed language communities [9]. As a result, there is a history of disenfranchisement which adds a barrier of distrust to the barrier of language that exists between deaf and hearing communities. At present, current technologies claiming to translate between spoken and signed languages are not viewed favorably by signed language communities. Rather, technology is often perceived as a ploy to replace human interpreters [10,11], or even as cultural appropriation by predominantly hearing researchers, who do not always have basic knowledge of these languages, and often have little connection with signed language communities [12]. Linguists have noted that as long as avatars are only capable of artificial and flawed language, they are very likely to be counterproductive [13].

This skepticism towards automatic translation and synthesis systems is exacerbated by the generally poor quality of their signed language [14]. To date, these have exhibited robotic movement and are mostly unable to reproduce all of the multi-modal articulation mechanisms necessary to be legible. Moreover, many of these systems are not yet able to take into account all of the linguistic phenomena specific to signed languages. This is analogous to early speech synthesis systems which produced robotic-sounding voices because wave forms were concatenated with little regard to coarticulation and no attention to prosody.

### 1.2. Language Quality

As with any machine translation system, users will judge the application by the correctness of the translation. The same is true when the target language is signed. Poor-quality signing is difficult to understand, just as poor-quality or egregious misspellings are difficult to understand. It undermines the viewer's confidence in the quality of the translation. Worse, poor-quality signing alienates the signed language community. Being forced to struggle with poor signing is no better than being forced to lip-read or use captions in a second language.

This is evidence that reconfirms a continuing disenfranchisement. For these reasons, the quality of the signed language display technology must be given the highest priority in a translation system. The motion should be indistinguishable from that of a human signing the same utterance. This should be the ultimate goal of any signed language display, because the brain is able to distinguish between biological and non-biological movement. If the movement is not identified as biological, this will undermine its acceptability as a human movement.

### 1.3. Challenges to Signed Language Display

Among the challenges to acceptable signed language display, three issues stand out. The first is understanding the differences between the modalities of signed and spoken languages, and the second is satisfying the requirements for developing the technology necessary to display signed languages. The third is the development of a representation that can act as the connection between the corresponding text of a spoken language and the geometry of the display technology. The quality of the target language will be determined by the underlying representation.

### 1.4. Modality

The modality of signed languages differs markedly from that of spoken languages, which utilize the vocal apparatus for production, and hearing for the reception. Spoken languages use visible communicative behaviors like gestures as well, but listeners can comprehend audio-only sources. In contrast, signed languages use visible actions for production, and vision for the reception. Whereas speech utilizes a single vibrating column of air for producing utterances, signed languages use spatial configuration, rhythm and speed of multiple body parts concurrently, including arms and mouthing as well as head, face, eyes, torso, and fingers on the hands.

All signed languages have linguistic processes that are not linearly ordered. For example, Figure 1 demonstrates the use of pursed lips to intensify the sign SMOOTH in American Sign Language (ASL). Layers of processes ranging from the phonological to the prosodic can co-occur in signed languages [15]. Co-occurrence is a more general term than synchronized or simultaneous, as co-occurring events do not necessarily start or end at the same time, but they overlap in their duration.

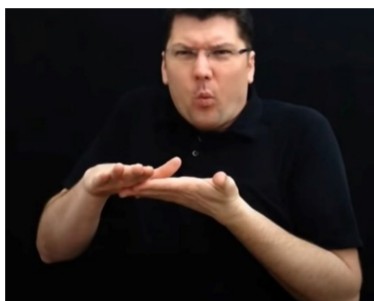

**Figure 1.** Pursed lips intensify the sign SMOOTH in ASL [16].

Although there are many discrete lexical items in signed languages, much information is conveyed through forms with infinite variability and depiction, unlike fixed dictionary signs. A case in point are classifiers in signed language, which represent general categories or "classes" of objects. They can be used to describe the size and shape of an object, and they can also represent how an object moves or is utilized. By using classifiers, a signer can describe a scenario with few discrete lexical items. The signer creates an image in space. This is not simply an informal gesture as there are well-documented linguistic rules governing classifier usage [17]. These are evocative, not necessarily iconic, and are extremely powerful. Dudis [18] analyses a narrative of a motorcycle ride, where a signer uses an instrument classifier to indicate that the rider is revving the engine and a vehicle classifier to show the rider driving away on a hilly highway (Figure 2).

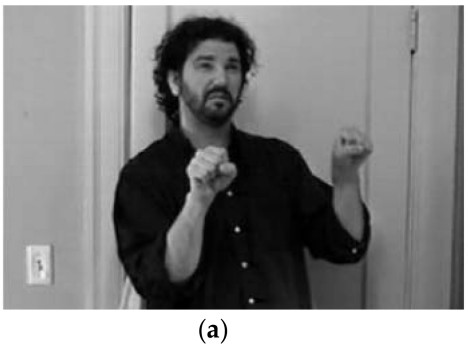 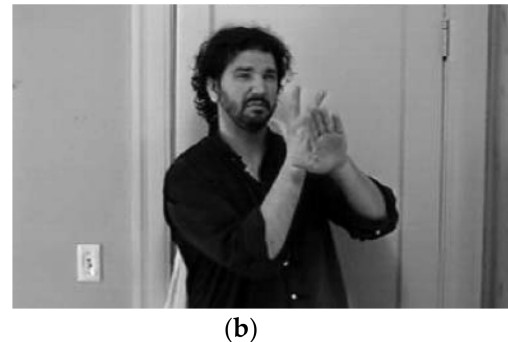

(**a**)                                                                                                   (**b**)

**Figure 2.** Classifier usage [18]. (**a**)The motorcyclist. (**b**) driving up a hill.

The presence of multiple articulators that can co-occur, the usage of classifiers, and the linguistic use of the signing space to organize the discourse are examples of the stark difference between signed and spoken languages. For these reasons, it is essential to avoid the trap of casting the problem of signed/spoken translation as a case of simply retrieving lexical items or phrasal units from a dictionary and concatenating them.

### 1.5. Display Technology

The second challenge is the development of technology capable of displaying signed language. Using the technology of three-dimensional (3D) avatars appears promising because it has the advantages of consistency and flexibility. When recording a human signer with traditional video, special care must be taken to ensure consistency of the studio setup and the appearance of the signer between recording sessions. This requires additional time and money. When using an avatar, the lighting and camera setup can be fixed; the clothing can be chosen by the viewer as can the hair and makeup. No additional resources are required to ensure consistency.

In addition, avatars have the advantage of flexibility through the use of animation techniques. They can display co-occurring linguistic processes. Proper application of coarticulation can provide smooth transitions and can inflect signs according to syntactic rules. These properties are necessary for a translation system to produce novel utterances. An in-depth history of past sign language avatar development is available in [3,19].

Avatars also have flexibility in appearance. They can be adapted to look like a specific person or a cartoon character. This flexibility in appearance can also anonymize a signer, so that the signer's identity will remain hidden.

Another advantage of this type of anonymization of content is that it covers one of the key properties of written language, which is inherently more anonymous than a live performance that is spoken or signed. With an anonymously presented avatar, content can be communicated without knowing the person who expressed it.

Given that there is a century's worth of development in animation, and nearly half that supporting video game technology, it would be tempting to dismiss the question of using avatars to display signed languages as a solved problem. However, a closer analysis shows that there are still significant challenges yet to be fully addressed [3,13,19].

Animation, the precursor to avatar technology, is powerfully communicative. Animation artists abstract and emphasize the salient features of a character for greater audience appeal and engagement. Simplification of a character's appearance is vital to maximizing emotional impact. This is the reason that the eyes of Disney cartoon characters are twice the size of those of a human and spaced more widely apart.

However, the requirements for signed language display are different from those for portraying cartoon characters. Beyond communicative power, the display of signed language requires precision. It must adhere more closely to physical reality. For example, the hands of animation characters such as Mickey Mouse or Homer Simpson have only three fingers. For a hearing audience, this is perfectly acceptable, but three fingers are not enough to distinguish between the fingerspelled letter W and the number 4 (Figure 3).

Another consideration is that while character animation effectively uses the face and body to express emotion, facial animation is typically of a lower quality than what would be required to portray a signed language legibly.

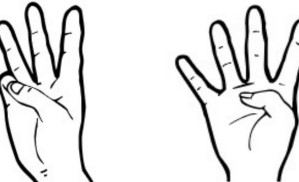

**Figure 3.** The difference between the letter W and the digit 4 would disappear in a three-fingered character.

Several ground-breaking animations have received attention and praise from signed language communities [20,21]. These were manually created by artists with the help of motion capture of human signers to extract a 3D skeleton. This approach enabled the portrayal of underlying natural processes of coordinated muscle action, coarticulation at a biomechanical level, and ambient movement. While creating the animation the artists are continually checking whether the animation draft effectively communicates the intended message and then editing the draft when there are flaws. However, animations are intended for playback only and are not extensible without manual intervention. Once completed, they are archived, and without additional manual editing cannot be utilized for generating new utterances. In short, new animations cannot be created in real-time and the approach is not interactive.

In contrast, video game characters move in response to player input in real-time and are highly interactive. Thus, using video game technology might seem like an expedient approach to signed language display for a translation system. However, many game players continue to comment on the poor quality of the game characters. This is due to the effect of the uncanny valley [22]. If a character appears more human-like, viewers expect the character to behave in a more human-like manner. But because the character's motion cannot be refined and edited by human animators before it is displayed, the results are unsatisfying. As explained by a professional animator [23]:

> For something like film or television, I could create a kickass animation of a monster jumping off a building and landing on the street below, but to do the same thing in a game, the movement has to be broken up into separate parts. This is because he probably won't do the exact same action every time. There may be buildings of different heights in the game, so I can't hard-code the height of the jump into the animation. I have to create an initial jump animation, then an idle hang-time animation to play while he's in the air, and then a landing animation. The programmer then strings the jump, hang-time, and landing together and decides the timing and trajectory of the hang-time part procedurally. That takes artistic control away from the animator and can result in some fugly animation.

Unfortunately, a "fugly" motion on a signed language avatar can destroy the legibility and even the meaning of the message, thus making the avatar bothersome or even useless for a deaf signed language user. Finally, the representation of signed languages through avatars will have an effect on the hearing perception of these minority languages. Hearing viewers should not be confronted with "fugly" signed texts and be misled into thinking that it is real signed language in all its beauty and richness.

The analysis of the requirements for a signed language avatar shows that it must have the expressivity of manual animation but the flexibility of an automated video game character. These two requirements are in conflict. It is still an open question as to how to reconcile these goals, but the most promising path forward is through a representation of signed language that an avatar can utilize for creating animations. There are many attributes to consider when choosing from current representations or when developing new ones.

The remainder of the paper discusses considerations for signed language representation and surveys the representations currently in use.

## 2. Representation Considerations

Avatars can be appealing in appearance but they must move naturally and correctly if they are to convey signed language that is acceptable and easy to understand. In other words, a signing avatar's quality is dependent on the directions it receives on how it should move. The directions for movement must come from a text-based representation that is also compatible with machine translation. Current automatic translation techniques largely require that the languages being processed have a written form. Signed languages do not have a written form. They are languages and cultures that have been preserved and transmitted from generation to generation by "hand to eye to hand". This section examines the features that are helpful for supporting acceptable avatar movement.

The features include text searchability, anonymity, full 3D information, motion, incorporation of nonmanual channels, level of detail, asynchrony, corpus support and ease of authoring. Many of these features first appeared in annotation systems for signed language analysis, but this paper examines their effectiveness for supporting signed language synthesis.

### 2.1. Digital Availability

This is a requisite for any type of automation. It facilitates such activities as searching, indexing, linguistic analysis, dictionary-building, and machine translation. Text-searchable representations have opened doorways to developing standards for information exchange, corpus building, and accessibility.

### 2.2. Anonymity

The goal of providing an option for anonymity online is to encourage self-expression. When writing under an unidentifiable pseudonym, authors can post comments on opinion-sharing websites without fear of backlash that could affect them in the physical world. But beyond the use of unidentifiable pseudonyms and circumventing tracking technology, there is another requisite to preserving anonymity, which is the use of text to post a comment. Leaving a video recording instead destroys the poster's anonymity. A positive attribute to a signed language representation would be the facility to preserve a signer's anonymity in the same way that typing a text message can preserve anonymity.

### 2.3. Specifying Full 3D Information

A human body exists in three dimensions, and a complete description in three dimensions can fully specify a human pose at any instant in time. The three dimensions can either be explicit or can be looked up through means of a library.

### 2.4. Specifying Motion

Signed languages utilize movement and spatial patterns to transmit information. In the early 1980s, Poizner [24] investigated the relationship between movement and the perception of language. To isolate movement from other aspects of a signed language, researchers attached small incandescent "grain of wheat" lights to the signer and videotaped the signer in a darkened room. By adjusting the brightness and contrast, only the moving spots of light were visible during playback. Signers observing the point-light displays were only slightly more accurate when viewing a well-lit recording. This is compelling evidence that the motion of signing should be part of a representation system.

### 2.5. Self-Evident

Is it possible for a person fluent in a signed language, to view symbols in a notation and recognize the sign? Is the notation simple enough that it does not make excessive demands on the reader? This would be analogous for a reader of a spoken language,

using knowledge of phonics, to "sound out" the symbols of a written word. In such a representation of a signed language, symbols, when viewed together as a whole, would reveal the identity of the word. This attribute facilitates manual review and editing, which is a valuable asset when building a corpus.

A second method that provides user-friendly access is to use multimedia. In this approach, the notation appears in a time-aligned format with a video recording [25,26].

### 2.6. Incorporates Nonmanual Channels

Although the formal linguistics of signed language began with an analysis of the manual channel, including hand shape, location, and orientation as well as movement, later research revealed that these were only a portion of the linguistic processes occurring in a signed language. The shoulders, spinal column, head, eye gaze, and face, including brows, lids, cheeks, and lips all participate in the production of such utterances as posing questions, expressing appositives, and introducing topicalization.

### 2.7. Level of Detail

To produce lifelike, legible utterances that are easy to read, an avatar needs a massive amount of detailed information. At the fundamental level of computer graphics technology, each frame of animation requires information specifying the orientation of each joint in the avatar. This requires a tremendous amount of numeric data. There are 206 joints in the human body and over 600 muscles [27]. A challenging aspect is determining an effective strategy for managing the data. The list that follows discusses possible options. The strategies listed progress from biomechanically- oriented approaches to those that focus on linguistic perspectives.

#### 2.7.1. Motion Capture and Video Tracing

Newer technologies have been able to measure and record data at such a high resolution that than can capture fine wrinkles and even skin texture (Figure 4). Commercial companies have used this successfully to create convincingly lifelike communication. Representatives of this technology encompass motion capture devices, including body suits, data gloves and facial tracking (Figure 5).

Over the years, mechanical sensors have largely been replaced with visual markers, which have become smaller and less physically constraining. This has been a good development for signed language recording as wearing gloves or a mocap suit affected the production of the signing [28]. Later innovations developed markerless technology, which relies on distinctive body features [29]. For example, markerless facial capture uses features such as the nostrils, and the corners of the lips and eyes.

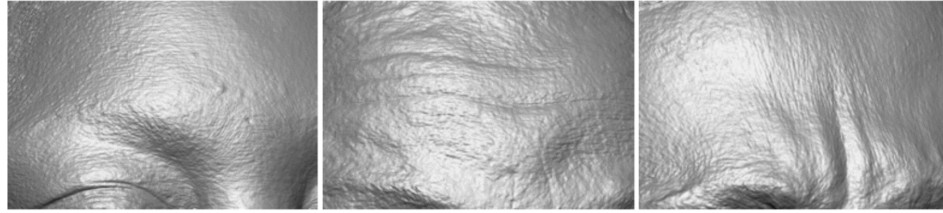

**Figure 4.** Fine detail recorded through capture technology [30].

As appealing as this appears, several aspects pose significant challenges. Captured data is noisy (full of errors) and requires a significant investment of skilled manual labor to clean up the data before it is usable. Further, if a signer's hand blocks the face, information about the face most likely will be missing, because a marker must be visible to the camera in order for the camera to record it. To counter this, many motion capture systems use multiple cameras. Such a setup can become prohibitively expensive.

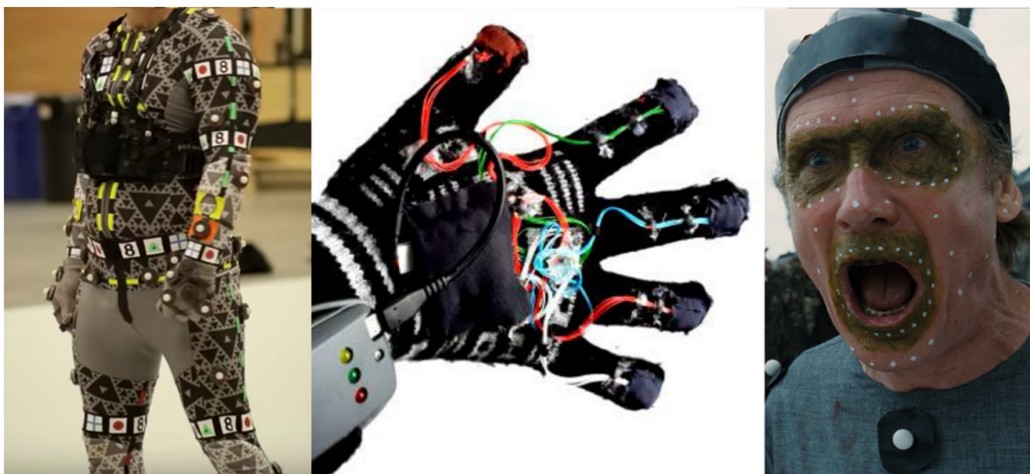

**Figure 5.** Motion capture suit [31], data glove [32] and markers for facial tracking [33].

A promising alternative is to glean motion data from video recordings. Researchers are developing tracing software similar to OpenPose to identify hand, body, and facial key points [34,35]. However, the data acquired in this way only contains two of the three necessary dimensions because the video is lacking depth information. Although recent progress has achieved an accuracy in depth estimation of 20 cm with a 95% confidence interval, this is not yet accurate enough for accurate sign language portrayal [36]. When relying on a single camera, these approaches are vulnerable to optical occlusion, for example when a hand hides the face, as demonstrated in Figure 6.

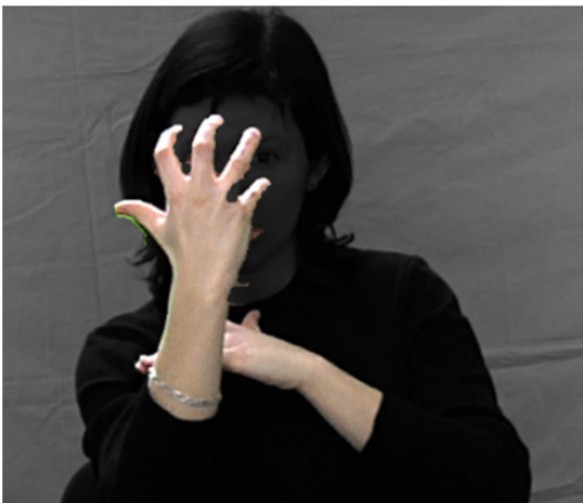

**Figure 6.** Facial occlusion can occur while signing [37].

Counterbalancing this drawback is the wealth of freely available video data from such sources as YouTube and Deaf social media [38]. The magnitude of this data far exceeds that of motion capture data. Further, the video is available at a sharply reduced cost, which is the compelling attraction to this approach. However, researchers need to ensure that all proper right-to-use stipulations are honored.

The animation created by these capture and tracing technologies can be quite convincing in playback, but because the techniques produce such an enormous amount of data, it is impractical to use the data for anything else. Attempting to divide captured data into segments for reassembly is time-consuming and the new productions created from the reassembled data are not satisfying to viewers. Editing the massive amounts of data manually is impractical, but researchers are looking for ways to automate editing, and the initial results are promising [39].

### 2.7.2. Key Frame Animation

Still biomechanical in perspective, but much easier to edit are approaches that use key frame representation. Instead of storing data for every joint for every frame, key frame approaches store data for a selected set of joints for a few, most characteristic ("key") frames of a sign. Figure 7 shows the two keyframes necessary for the sign AGAIN in ASL. This approach requires less than one percent of the storage resources required for mocap data. Because there is less data, it is also easier to edit to support language processes that require changes to a sign's form. Using key frames from a sign library can create naturally appearing animation that is easy to read [40].

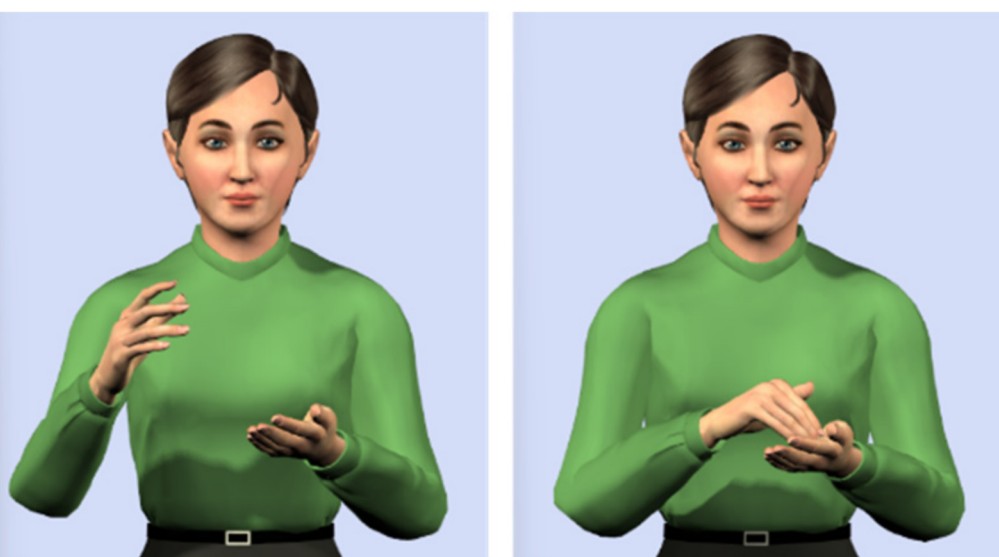

**Figure 7.** Two key frames define the sign AGAIN [38].

The disadvantage of this method is its reliance on manual labor. Creating convincing signs using key frame animation requires the same skill set as is needed to create Disney movies. Finding qualified animators can be a challenge.

### 2.7.3. Phonemic and Phonetic Representation

Although the capture and key frame approaches have the potential for producing realistic detail, they do not convey linguistic information. The streams of numbers from capture or tracing do not contain insights into the linguistic processes that are occurring. In contrast, a parametric representation system that capitalizes over a half-century of linguistic research can give insights into the language processes that are present in an utterance. Stokoe's [41] pioneering work identified three manual parameters, namely hand shape, location, and movement. To these three, Battison [42] added palm orientation, and, subsequently, other researchers [43,44] identified non-manual features as a fifth parameter. Figure 8 gives examples of each parameter. Each row shows a minimal pair: two signs that differ only in the parameter being demonstrated.

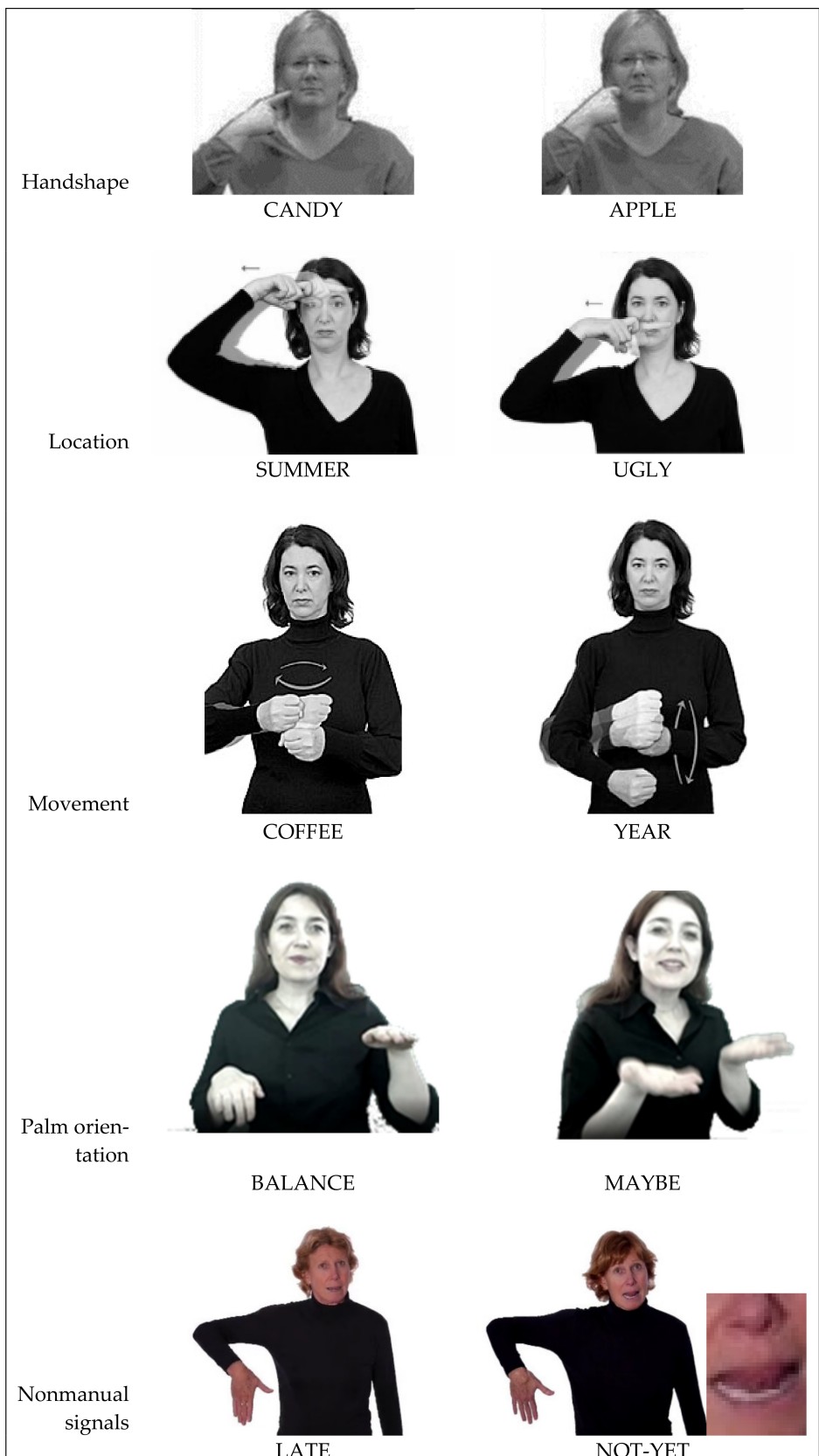

**Figure 8.** Five phonemic parameters of signed language. All examples are in ASL [45–48].

Phonemic classifications are specific to a particular language. For example, researchers have identified different sets of handshapes in French Sign Language (LSF) and

ASL [41,43,49,50]. The need became apparent for a system that could accommodate the representation of any signed language, regardless of geographic location. Such a system would be analogous to the International Phonetic Alphabet, whose aim is to provide a unique symbol for each sound in a language and to serve as a standard for transcribing spoken language. To this end, researchers have been developing and utilizing phonetic representations since the 1980s [51–53].

### 2.7.4. Lexical and Morphological Considerations

In published articles on signed language linguistics, a transcription of a signed language utterance often appears as a series of glosses, where a gloss is a word or words in a spoken language that most closely correspond to the meaning of the sign in its citation form. However, simply listing glosses in order that signs occur will give little indication of a sign's form, because the glosses designate signs in their citation forms [54]. Signed languages exhibit rich morphological structures, including verb agreement, classifier depictions, and verbal aspects. For example, the verb GIVE in ASL can be inflected for subject and object agreement, number, temporal aspect, and can be accompanied by a grammatical facial nonmanual that functions as an adverbial modifier [24,55]. Researchers have developed multiple conventions for labeling morphological processes [52,56,57]. Some use indices that appear alongside a gloss to indicate relationships with other elements in the utterance, or a short descriptor explaining a change from citation form (Figure 9). Although there are similarities among many of the systems, no single standard has yet emerged.

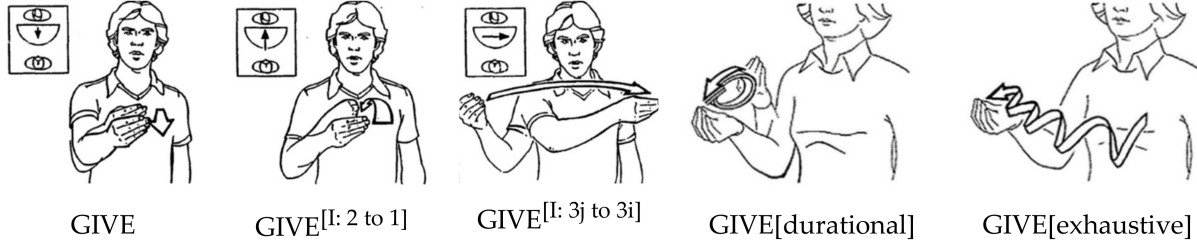

**Figure 9.** Notations for several inflections of GIVE [58,59].

### 2.7.5. Phrasal Level

Drawing from written forms of spoken languages, annotation systems for sign transcription often use commas to delineate phrases and periods or question marks to terminate sentences [60]. Because signed languages use nonmanual signals to mark topics, rhetorical questions, and conditionals, a common practice is to draw a line above the clause, together with a label to designate the type of clause as seen in Figure 10a.

<div style="text-align:center">

<u>top</u> ________________<u>pol-q</u>
CAT,  INDEX$_a$ ALLERGIC?

(**a**)

____<u>br</u> ________________________<u>bf</u>
BOOK, YOU WANT WH-MANY

(**b**)

</div>

**Figure 10.** Notation strategies for phrases. (**a**) As for cats, is s/he allergic to them? (**b**) As for books, how many do you want?

The convention varies. In some situations, the label designates the physical manifestation such as brow posture or body leans, as seen in Figure 10b [61].

### 2.8. Asynchrony

As mentioned previously, one of the intriguing aspects of signed languages is their use of multiple articulators. Deaf composer Christine Sun Kim describes this nonlinearity in her TED talk [62] where she draws on music as a metaphor to explain this aspect of signed languages. She likens spoken language to a single note, and signed language, with its multiple articulators, is like playing a chord. Expanding on this metaphor, spoken

language is a melody line played on a solo instrument. In contrast, a signed language, with its multiple articulators, is an orchestra with many interconnected and cooperating parts contributing to the rich texture of the performance.

However, this metaphor breaks down in that the timing in an orchestral score is too constraining for signed languages. In a musical work, all activity is synchronized to a beat. This rhythmic straitjacket is visually apparent in any music score through the typesetter's selection and placement of musical symbols. Signed language is not constrained by this type of synchrony. Changes in sign parameters are not limited to the beginning or conclusion of a sign. In fact, when an orderly synchrony, so crucial to ensemble music performance, is applied to avatar movement, the result is the halting, robotic awkwardness universally decried by Deaf viewers.

*2.9. Corpora Availability*

A formal definition of a representation is a necessary beginning, but in order for it to be useful, there must be data available that demonstrates the representation. Just as with spoken languages, corpora are essential for the automatic production of language. They form the basis for gaining insights into the structure and usage of a language. They can serve as the ground truth for animations produced by an avatar. Signed language corpora are multimodal; they contain digitized video recordings that are time-aligned with annotations [26,63]. The annotations may contain multiple descriptors, including a translation into a spoken language, glosses corresponding to the signs, and/or a notation specifically designed to represent sign structure. Annotations must be added manually, and the labor costs for annotation and cross-checking are contributing factors to the scarcity of corpus resources.

*2.10. Easy Authoring*

Certainly, an important purpose for a signed language representation is to serve as a target in automatic machine translation. However, some applications, such as the creation of educational materials, are best facilitated by manual authoring of signed prose [64]. Such applications require more than individual signs performed in isolation and they benefit from a representation that facilitates quick and easy authoring. Hand-in-hand with easy authoring would be the prospect of easy editing. If an author views the utterance signed by an avatar and wants changes, the goal would be an approach that makes sign editing as easy as text editing in current word processors.

## 3. Comparing Representations

The rows in Table 1 list technologies and research efforts to represent signed language, and the columns contain the useful qualities for supporting signing avatars discussed in Section 2. The sources given are not exhaustive but are a good starting point to explore each type of technology. Several systems (Stokoe/ASCII Stokoe; HamNoSys/SiGML) appear as pairs. The first item in each pair is a linguistically based notation system and the second item is a computer-readable specification that supports the notation.

The ASCII Stokoe system represents the symbols of the Stokoe phonemic notation as ASCII characters, which made the notation compatible with computer processing. Grieve-Smith took advantage of this to develop an early sign authoring system as a Web-based avatar display [65]. In contrast, the goal of SignWriting notation was not to facilitate linguistic analysis, but to provide a writing system for signed languages. Researchers are developing an XML-compliant format called SignWriting Markup Language (SWML) to aid in processing SignWriting texts and creating dictionaries. In addition, efforts are underway to use SWML to drive an avatar display [66].

HamNoSys is a phonetic transcription system for all signed languages whose purpose is analogous to that of the International Phonetic Alphabet for transcribing spoken language. Although HamNoSys is machine-readable, researchers developed SiGML, Signing Gesture Markup Language, as an XML-compliant format that is "more amenable to computer



processing" [67]. The purpose of SiGML is to support signed language generation through avatars, and so required additional information beyond the data required for linguistic analysis. To the original specification of SiGML, Glauert and Elliott [68] added a framework that specifies timings internal to a sign. The framework incorporates Johnson and Liddell's SLPA model which decomposes signs into a series of consecutive segments, each of which lists any changes occurring to articulators during the segment [69]. An advantage of this approach was that changes to articulators were no longer constrained to sign boundaries but were constrained to segment boundaries. Several avatar-based applications have utilized SiGML or HamNoSys including the eSign Editor [70], the ViSiCAST animation component [67], and JASigning [71].

**Table 1.** A comparison of representations.

| Representation | Digital Availability | Anonymity | Full 3D | Specifies Motion | Self-Evident | Non Manuals | Asynchrony | Level of Detail | Corpora | Easy Authoring |
|---|---|---|---|---|---|---|---|---|---|---|
| Line drawings [1] | | yes | | | | yes | | Lexical | | |
| Video recordings | | | | yes | yes | yes | | | | |
| Motion capture [2] | | | yes | yes | | see [3] | yes | Fine detail | | |
| Gloss | yes | yes | | | | | | Lexical | see [4] | yes |
| Stokoe/ ASCII Stokoe | yes | yes | | | | | | Phonemic | see [5] | |
| HamNoSys/SiGML | yes | yes | yes | see [6] | | yes | | Phonetic | see [7] | |
| SignWriting/SWML | yes | yes | | | yes | yes | | Phonetic | see [8] | yes |
| SLPA model/SiGML [9] | yes | yes | yes | yes | | | | Phonetic | | |
| Berkeley Transcription System [10] | | yes | | yes | | yes | | Morphological | | |
| Qualgest [11] | yes | yes | yes | yes | | | | Phonemic | | |
| EMBRscript [12] | | yes | yes | yes | | yes | yes | Key frame | | |
| Zebedee [13] | yes | yes | yes | yes | | | | Phonetic → Morphological | | |
| AZee [14] | yes | yes | yes | yes | | yes | yes | Phonetic → Phrasal | see [15] | |

[1]: [74–76]. [2]: [77,78]. [3]: Motion capture suits record nonfacial, nonmanual signals. Recording facial nonmanual signals requires additional equipment. [4]: Usually, glosses are part of a representational system that annotates more than the citation form of the sign and are part of a multimedia corpus. See [25,26,52,79]. [5]: Irving [80] describes a project to encode 5000 ASL signs as ASCII-Stokoe for synthesis. [6]: HamNoSys can specify temporal ordering of signs and articulator locations within a sign. [7]: Many corpora utilize HamNoSys annotations. See [25,26,79]. [8]: Available at https://sigbank.org (5 April 2022). [9]: [53] describe efforts to incorporate SLPA [67] into a phonological corpus tool. [10]: [56]. [11]: [81]. [12]: [82]. [13]: [72]. [14]: [73]. [15]: Rosetta: https://www.ortolang.fr/market/corpora/rosetta-lsf (5 April 2022) AZee descriptions and annotations of each video and 40brèves: https://www.ortolang.fr/market/corpora/40-breves (5 April 2022).

Several of the representations address the morphological and/or phrasal levels of language. The goal of the Berkeley Translation System was to facilitate analysis of adult-child interactions in signed language rather than as a support for signed language generation [54]. In contrast, the Zebedee model supports morphological processes through geometric constraints and parametrizable scripts [72]. A subsequent method, AZee, facilitates a more finely detailed and realistic timing of articulator changes, couched in linguistic terms [73].

## 4. Conclusions and Future Directions

As can be seen in Table 1, no one approach has all of the features for a representation that completely supports signed language synthesis by avatar technology. This is understandable as some of them are in direct conflict with each other. How can a representation have a written form that lends itself to quick authoring and easy editing but also contain a sufficient level of detail that an avatar system has enough information to create motion that is legible? How can a linguistic abstraction be realized geometrically in animations that are easy to read? How is it possible to automate processes that currently require manual intervention? Answering any one of these questions will propel the state of the art forward to bring us one step close to the ideal of an automatic spoken-to-signed translation system that is acceptable to the Deaf community.

**Funding:** This research was funded in part by the EASIER (Intelligent Automatic Sign Language Translation) Project. EASIER has received funding from the European Union's Horizon 2020 research and innovation programme, grant agreement n° 101016982.

**Institutional Review Board Statement:** Not applicable.

**Informed Consent Statement:** Not applicable.

**Data Availability Statement:** Not applicable.

**Acknowledgments:** The authors thank the reviewers for their thorough reading and helpful comments.

**Conflicts of Interest:** The authors declare no conflict of interest. The funders had no role in the design of the study; in the collection, analyses, or interpretation of data; in the writing of the manuscript, or in the decision to publish the results.

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
