# Peer review of "Sign Language Avatars: A Question of Representation"

_information, doi:10.3390/info13040206_

Round 1
Reviewer 1 Report
The paper is devoted to many aspects that are associated with the recognition of both individual gestures and sign languages. The authors of the paper describe in detail the main problems that arise in the process of solving problems related to automatic machine sign language translation. The paper clearly shows that despite the great practical potential, the problem of effective sign language recognition has not yet been solved due to serious differences in the semantic and syntactic structure of written and sign languages, as a result of which it is not yet possible to perform an unambiguous translation from a sign language into, for example, a text language representation. Therefore, there are currently no operating fully automated models and methods for sign language translation systems. To create such full-fledged models, it is necessary to carry out a deep semantic analysis and analysis of written phrases, and this is still possible only at a superficial level due to the imperfection of text analysis algorithms, knowledge bases, and avatars. It is also well noted that the hearing impaired are limited in their ability to communicate with hearing people, and when applying to various government agencies, they are sometimes provided with sign language interpreters, which are often not enough. However, in my humble opinion, the paper is not free from a number of shortcomings. First of all, it is striking that the authors quite extensively describe the problems of machine sign language translation, but there is little description of the problem that arises, among other things, due to the lack of universal methods for creating multimodal sign corpora, as well as methods and algorithms that improve the efficiency of machine learning and accuracy of automatic recognition of sign languages using various video information capture devices that allow obtaining not only high-quality images in the optical mode, but also additional data on the coordinates of graphic areas of interest (depth map mode, infrared mode, etc.). After all, on the basis of such universal methods for creating multimodal gestural corpora, it is possible to create corpora that can later be used, among other things, for the analysis of a non-verbal way of communication (transmission of thoughts, feelings and emotions of a person through body movements and hand gestures) and this can lead to the creation of more realistic avatars. Thus, it may be worth expanding the paper by analyzing a table with already available gesture corpora (InterHand2.6M, TheRuSLan, WLASL, AUTSL, ChaLearn Synthetic Hand Dataset and others). Also, to the described OpenPose library, it is worth mentioning the approach for determining the orientation of human hands, which is implemented in the open-source cross-platform environment MediaPipe, and perhaps, about a group of researchers from the Facebook AI Research center who are developing the FrankMocap library, focused on two-dimensional tracking of all parts of the human body, including the areas of the hands, with their further three-dimensional visualization. It seems to me that all the proposed additions will only improve this paper, and it will be useful and interesting to many. Finally, the style of the paper requires minor revision due to the presence of spelling and punctuation errors.
In general, the paper should be finalized and then it can be accepted for publication.
Author Response
Please see attached pdf file.

Reviewer 2 Report
Dear authors. Foremost, we want to share our appreciation to your work in means of its social impact on deaf community. This is a very important part in making a society without borders, where people with disabilities (hearing disabilities in your case) can gain access to all benefits of community and possibility of communication with other people and deaf people in particular. However, besides social impact of your article, there are issues in your article that has to be fixed. Also, there are some suggestions for your consideration. For your pleasure they will be split in two parts:
Suggestions:
- In part Abstract, we suggest avoiding phrase that “sign languages don’t have written form” and rewriting it in such manner “sign languages do not have standardized written forms” since there are sign language notation systems (Stokoe notation, HamNoSys, SignWriting etc.)
- In part Abstract, we suggest adding disambiguation for word “avatar”, since for people which are not familiar with sign language, this word can be mistaken with animation film; in such case we suggest using such phrase “3d animated human model (avatar)”
- On line 23 we suggest adding “graphic and symbolic” after word “textual” since sign languages can be depicted using symbols, glyphs, images, pictograms, glosses etc.
- In part Keywords we suggest adding keyword “computer animation” since sign language avatars can be treated as computer animated models and can wide range of readers of this article
- In part Introduction, on line the sentence contain word “orthographic”, but since the article is dedicated to sign languages, this sentence needs some additional remark, e.g “spoken written languages”
- On line 41 we suggest adding first reference to article/book dedicated to Stokoe notation
- On line 42 we suggest noting the difference between spoken and signed languages, using terms “sequential”and “parallel”.
- On line 59, we suggest re-write the sentence, using such phrase “… institutions … don’t represent people from the deaf community” since it’s mostly the case and deaf communities are often isolated from the hearing community.
- On line 66-67, we suggest using phrase “are not familiar with”, since it means that people, noted in that sentence, don’t know basics of the sign language, its history, its structure etc.
- On line 68 we suggest adding footnote “with exception of usage of performance motion capture” since this technology aids researchers to recreate precise movements of human body and every aspect of gesture (location, hand shape, hand and body movements etc).
- On line 71 we suggest avoiding term “robotic”. This term is very subjective and it should have been replaced with term “synthetic”, since movements in sign language, spoken by deaf signers can be percepted by hearing people like “robotic” because of exteme speed of speech. And in contrary, very slow and calm movements can be incorrect by nature and treated by signers like “wrong” and containing serious mistakes in appearance.
- On line 74, we suggest using phrase “they repeat path/development of”, since development of every computer system can find analogy or counterpart in another.
- On line 87 we suggest using synonymous world “visual animation” for “display” in order to not being confused with computer components (display) and it’s more self-explanatory term.
- On line 88 we also suggest replacing term “indistinguashable” with “understandable” since common computer animation technology and virtual reality are still treated by users as virtual and synthetic (not natural).
- On line 90 we suggest adding footnote: “with exception of motion capture technology”.
- On line 99 we suggest adding remark or explanation for word geometry which can be confused with either science or geometry of 3D scene. In this sentence it should be replaced with phrase “geometric shape and movement of synthesized gestures”.
- On line 120 we suggest adding remark for word “classifiers”. In computer science it’s known as a specific range of algorithms. In linguistic sence it’s term depicting classes of signed words, e.g. “furniture”, “trees”, “animals” and unknown for people who are not familiar with sign language.
- On line 120 we suggest renaming the cited text as “sentence”, “conversation” as a linguistic phenomena, since “story” is more common word term and less desirable in context.
- On line 129 we suggest naming the articulators of the sign language in brackets or adding disambiguation of the word.
- On line 144 we suggest adding examples (references) to known sign language avatars in order to improve the readers’ knowledge about the avatar technology. Same can be said about the line 148, where the authors note the “cartoon” sign language avatars which represent development of the avatar technology.
- On line 149 the authors underscore the possibility for the signers to be hidden, but in contrary, the benefit of this is possibility to capture movements of different signers. In standard situations, the signers show the signs of fatigue and visual quality of the gestures become more “slower” and “less precise” than in the beginning of the session. Same can be said about the possibility to hide the author; the signers, like writers or artists have their own “style” and lexicon (knowledge base). Hence, some signers can be figured out, if their movements were captured using motion capture technology.
- On line 283 one should note only that systems that are usable in signing.
Issues:
- Line 33 and further (for instance, line 55) consists incorrect reference symbol “()”. Use “[]” while citing a source instead since “()” is used for mathematical formulae and theorems.
- Sentence on line 43 repeats sentence in abstract on line 19; same can be said about line 225, where the sentence repeats another sentence on line 18. In some cases this is acceptable while writing separate articles (so-called reutilization of scientific results), but in case of scientific article we suggest rewriting such sentences to avoid unnecessary repetitions.
- On line 47 we suggest replacing term “desirable features” with term “significant features” since features in context, which is mathematical and scientific, are mostly descriptive and not represent any affect, emotion or feeling
- On line 79 we suggest using more common or self-explanatory terms; hence, it’s better to replace terms “quality” with “usability”, “output” with “representation” or “translation”.
- We suggest avoiding “emotional” words and phrased in scientific articles dedicated to computer technology or science, for instance “bad”, “good”, “sad”, “feels” and others, with exception of articles dedicated to human emotions and human psychology; for instance, word “poor” in context of computer technology, used on line 81 is very subjective and represent feeling about it, but does not represent it’s essential features – “quality”, “accuracy”, “level of detail”, “performance”. Same can be said about sentence on line 84 – “being forced” is dedicated to articles in “law” category. In context of this article it’s rather good idea to use more neutral phrase “deaf people in everyday life are coping with problem of using foreign (spoken) language” which is the case in this sentence. Also, on line 220 word “beautiful” can be replaced with more neutral term “realistic”
- On line 89 we suggest removing footnote to Turing test since text-to-speech and text-to-sign technologies only perform direct translation from one language to another and can’t act as virtual system (e.g virtual friend or virtual assistant). One can argue that big language models like OpenAI GPT-3 can have very slight signs of “intelligence” and may pass Turing test with precautions, but same sign language models are unknown for present time.
- If the peer-reviewed article depicts original PDF file (text, manuscript), there issues on lines 128 and 169, since the images in original test should be placed on the same page as the citations (references) to these images. In case if the original text does not contain the issue, we strongly suggest following this recommendation.
- Sentence on line 158 needs at least one reference. Same can be said about lines 255 and 273, which contain some statements that need at least one reference to an article or book.
- Line 193 contains too long citation and has to rewritten in more compact form like: “as author … says about the … there are some issues is …, hence the … technology has some restrictions”.
- Line 231 notes the existence of non-manual channels in sign language but does not list them as follows; for instance, they may contain body movements, head orientation, facial expression.
According to our impressions about the article, the authors are very familiar with the topic, however, there are numerous suggestions that should be seriously taken in account by authors in future article redactions; also there are 10 serious issues, that should be taken in account before publishing of the article. From our view, the paragraphs in introductory chapter before 1.4 (according to list of suggestions) need serious rework before being published. We also consider, if possible, adding a least 1 paragraph (before “Conclusions” part) dedicated to results and advances made by authors in this area since the authors took part in “EASIER (Intelligent Automatic Sign Language Translation) Project”, and from the literature sources listed below the article already have numerous results in area of sign language notation (at least 20 years in row), sign language 3d animation and other results dedicated to the topic of the article.
Author Response
Please see the attached PDF file.

Round 2
Reviewer 2 Report
Thank you for your correct understanding of the comments and suggestions. I think this article has become clearer.